# Blood Group Interpretation Algorithm Based on Improved AlexNet

Ranxin Shen [1,2] , Jiayi Wen [1,2] and Peiyi Zhu [1,*]

1  School of Electric and Automatic Engineering, Changshu Institute of Technology, Suzhou 215500, China; shenrx1015@foxmail.com (R.S.); fitter1999@foxmail.com (J.W.)
2  School of Electrical Engineering, Yancheng Institute of Technology, Yancheng 224002, China
*  Correspondence: zhupy@cslg.edu.cn

**Abstract:** Traditional blood group interpretation technology has poor detection efficiency and interpretation accuracy in the face of complex conditions in clinical environments. In order to improve the interpretation accuracy of the automatic blood group interpretation system, the important role of deep learning in the blood group interpretation system was studied. Based on the AlexNet network model, this paper proposes an improved scheme because of its advantages in terms of speeding up the convergence training speed and enhancing the model's generalizability. However, it still needs improvement in terms of blood group interpretation accuracy. The improved AlexNet network model proposed in this paper added an attention mechanism to the network structure, optimized the loss function in the training algorithm, and adjusted the learning rate attenuation function. The experiments showed that compared with the accuracy of the AlexNet model, its training effect was remarkable, with an accuracy of 96.9%—an increase of 3%. Moreover, the improved network model paid more attention to fine-grained classification, minimized the loss rate, and improved the accuracy of system interpretation.

**Keywords:** blood type classification; AlexNet model; attention mechanism; loss function optimization; learning rate decay function; deep learning; microcolumn gel card

## 1. Introduction

Recently, in the field of blood type detection, the fully automatic blood group analysis instrument has long become a hot topic [1–3]. With the continuous development of clinical science and in-depth and persistent research studies on blood type, there are various methods and forms of blood type detection. Detection methods of automatic blood group analysis instruments have covered the international mainstream detection methods. The mainstream methods are such as the microplate method and microcolumn gel method [4]. The main directions that the blood group analysis instrument can detect roughly include ABO blood type analysis, Rh blood type analysis, irregular antibody screening, cross-blood matching test [5,6], etc.

The object of the subject points to the research study of the fully automatic blood group interpretation system based on the microcolumn gel detection method (Card-type) in the blood group analysis instrument. The automatic blood group analysis instrument is based on microcolumn gel detection technology (MGDT), with a microcolumn gel card as the experimental carrier, also known as a card-type blood group analysis instrument. The card-type blood group analysis instrument has the advantages of simple operation, high precision, high sensitivity, fewer specimens, long-term preservation, standardized results, etc. It is suitable for hospitals, disease control, entry-exit for the Inspection and Quarantine Bureau, and other medical institutions [7].

The most important part of the blood type interpretation system is the best training model parameters that the model loads. The best training model parameters are the final

derived model parameters by pretraining a large number of blood group image datasets. At present, the most mainstream training and learning method is deep learning. Deep learning is a kind of machine learning which is the essential path to achieving artificial intelligence [8–10]. Deep learning is studied to build neural networks that mimic the human brain for data analysis and learning features of datasets, such as images, text, and sound [11].

The fully automatic blood group interpretation system often uses the classification-based deep neural network model. It needs to learn various red blood cell agglutination phenomena and reclassify them [9]. With the development of the classification-based deep neural network model of today, there have been many excellent neural network structures. From the LeNet network model published in 1998 to the AlexNet network model in 2012, the VGG network model in 2014, and the ResNet residual network model in 2015, etc., countless research scholars have been innovating and developing deep neural network models suitable for various fields constantly [12–14].

The project chooses to innovate based on the AlexNet network model and applies it to the field of blood type detection by the microcolumn gel method to improve the reading accuracy of the existing blood group interpretation system.

## 2. Materials and Methods

### 2.1. The AlexNet Architecture

AlexNet uses an eight-layer convolutional neural network. The first five layers are the convolutional layers, and the last three are the fully connected layers in the neural network. AlexNet won the 2012 ImageNet Image Recognition Challenge by a substantial margin [15] and had input pictures sized at $224 \times 224 \times 3$. After the transformation of five convolutional layers, three max-pooling layers and activation functions to map the raw data to the hidden layer feature space are created. Finally, three fully connected layers act as "classifiers", dividing the raw data into multiple categories [16–18]. The AlexNet network model concept is shown in Table 1.

**Table 1.** The AlexNet Network Concept Table.

| Layer Name | Kernel Size | Kernel Num | Stride | Padding | Input Size | Output Size |
|---|---|---|---|---|---|---|
| Conv1/Relu1 | $11 \times 11$ | 96 | 4 | [1, 2] | $224 \times 224 \times 3$ | $55 \times 55 \times 96$ |
| LRN1 | / | / | / | / | $55 \times 55 \times 96$ | $55 \times 55 \times 96$ |
| Maxpool1 | $3 \times 3$ | / | 2 | 0 | $55 \times 55 \times 96$ | $27 \times 27 \times 96$ |
| Conv2/Relu2 | $5 \times 5$ | 256 | 1 | [2, 2] | $27 \times 27 \times 96$ | $27 \times 27 \times 256$ |
| LRN2 | / | / | / | / | $27 \times 27 \times 256$ | $27 \times 27 \times 256$ |
| Maxpool2 | $3 \times 3$ | / | 2 | 0 | $27 \times 27 \times 256$ | $13 \times 13 \times 256$ |
| Conv3/Relu3 | $3 \times 3$ | 384 | 1 | [1, 1] | $13 \times 13 \times 256$ | $13 \times 13 \times 384$ |
| Conv4/Relu4 | $3 \times 3$ | 384 | 1 | [1, 1] | $13 \times 13 \times 384$ | $13 \times 13 \times 384$ |
| Conv5/Relu5 | $3 \times 3$ | 256 | 1 | [1, 1] | $13 \times 13 \times 384$ | $13 \times 13 \times 256$ |
| Maxpool3 | $3 \times 3$ | / | 2 | 0 | $13 \times 13 \times 256$ | $6 \times 6 \times 256$ |
| FC1/Relu6 | 4096 | / | / | / | $6 \times 6 \times 256$ | 4096 |
| Drop6 | / | / | / | / | 4096 | 4096 |
| FC2/Relu7 | 4096 | / | / | / | 4096 | 4096 |
| Drop7 | / | / | / | / | 4096 | 4096 |
| FC3 | 9 | / | / | / | 4096 | 9 |

The ReLU function is used as the activation function in the AlexNet network model, which replaces the traditional Sigmoid and Tanh activation functions and successfully solves the gradient dispersion problem of Sigmoid when the network is deeper. Meanwhile, the LRN function is used to normalize the partial features. It uses the result as the input of the ReLU activation function, which can effectively reduce the error rate. Moreover, the network selectively ignores individual neurons in training using the random discard technique (Dropout). It is employed in the last few fully connected layers, which can avoid the overfitting of the model. The pooling layer uses an overlapping maximum

pooling layer. That is, the pooling range is larger than the step size, and it can avoid the averaging effect of average pooling. However, the disadvantage is that the calculations will become more difficult as the network deepens, which leads to the "degradation" of the network [16,19,20].

*2.2. Improvement Scheme Based on AlexNet*

The scheme is an improvement based on the original AlexNet network structure. In the structure, add the channel attention mechanism (SEBlock) between the first convolutional layer and the first max-pooling layer. The SEBlock will give each channel a weight so that different channels have different forces on the results. It is conducive to expanding the impact of available features on the results. In the training environment, optimize the cross-entropy loss function algorithm. It is conducive to speeding up the loss rate convergence. Meanwhile, the learning rate fixed step decay strategy is tuned to find the optimal learning rate. It is beneficial to improve the impact of the learning rate on the results.

2.2.1. The Channel Attention Mechanism SEBlock

The Attention Mechanism is a data processing method in machine learning, which is used widely in different types of machine learning tasks, such as speech recognition, image recognition, and natural language processing. There are currently two main types of attention mechanisms: the spatial attention mechanism and the channel attention mechanism [20].

When inputting an image, the neural network will extract the image features, and each layer has a feature map of different sizes. Among them, the common matrix shape of the feature map is [C, H, W]. When the model is training, the matrix shape of the feature map is [B, C, H, W]. Where B means the batch size, C implies the number of channels, H indicates the height of the feature map, and W denotes the weight of the feature map. When the network extracts the image feature layers, the ability of the network to extract features can enhance by adding the spatial attention mechanism or channel attention mechanism between the convolutional layers. When writing the code, it considers the attention mechanism between feature maps, so the input of the code is the feature map with shape [B, C, H, W], and the output is still the feature map with shape [B, C, H, W] [3].

The SEBlock proposed in the improved scheme is a channel attention mechanism. It adaptively recalibrates channel feature responses by explicitly building the interdependencies between channels. These modules can be stacked together to form an MLP network structure and generalized effectively over multiple datasets. The SEBlock module structure is shown in Figure 1.

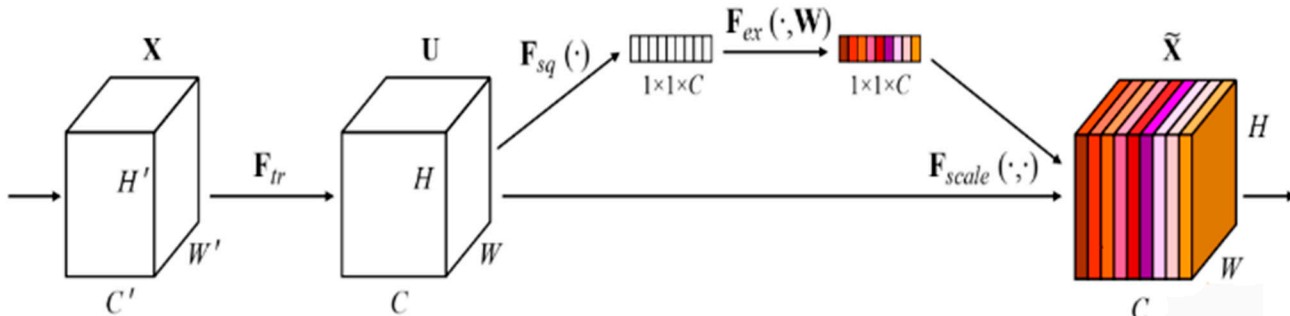

**Figure 1.** The SEBlock module structure.

The SEBlock module has four main steps: First, starting from a single image, the image features are extracted, and the current feature map dimension of the feature layer U is [C, H, W]. Second, average pooling or maximum pooling for the [H, W] dimensions of the feature map. The size of the pooled feature map is from [C, H, W] to [C, 1, 1]. [C, 1, 1] can be interpreted as for each channel C, there is a number corresponding to it one by one.

Third, the weights are extracted from each channel itself. The weights indicate the influence of each channel on feature extraction. After global pooling through the MLP network, the vector means that it obtains the weights of each channel. Finally, the weights obtained for each channel C [C, 1, 1] apply to the feature map U [C, H, W] [21]. That is, each channel multiplies its respective weight. The flow chart is shown in Figure 2.

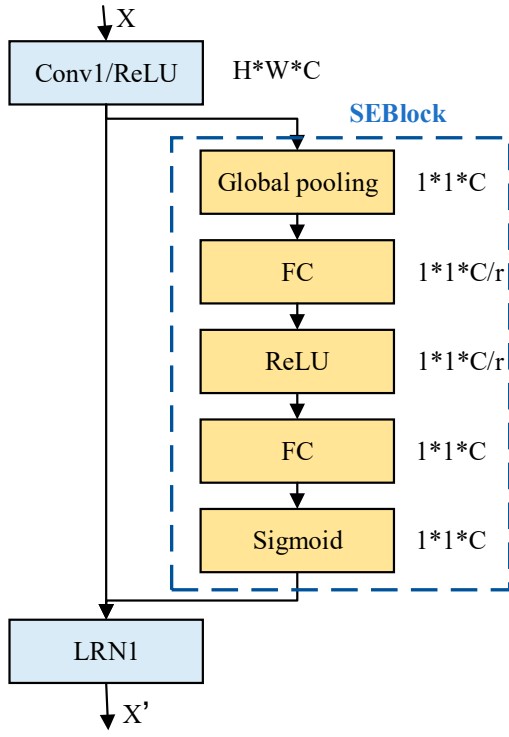

**Figure 2.** The SEBlock module step flow chart.

In this paper, after comparison experiments, it was found that the best result is when r takes 16, so the default r = 16. SEBlock is an MLP network consisting of an adaptive mean pooling layer and two fully connected layers, where the data are extracted from each channel itself via the MLP network with weight. The weights represent the influence of each channel on feature extraction [22]. The meaning of the globally pooled vectors after passing through the MLP network is that it obtains the weight of each channel, allowing the feature maps with a significant role to have slightly more influence on the results [3,21].

2.2.2. The Multi-Categorical Cross-Entropy Loss Function Algorithm Optimization

Cross-entropy loss is also known as log-likelihood loss and log loss. The cross-entropy loss function is used for multi-category tasks in deep learning where the model does not give the labels of the samples directly for the input sample z, but the probabilities of the labels. In multi-category tasks, the Softmax activation function combined with the cross-entropy loss function is usually used because cross entropy describes the difference between two probability distributions. However, the output of a neural network is a vector and not the form of a probability distribution. Therefore, the Softmax activation function is needed to "normalize" a vector into the form of a probability distribution [21,23]. Then the cross-entropy loss function is used to calculate the loss.

The multi-categorical cross-entropy loss function is calculated as follows:

$$
\begin{aligned}
Loss(x, class) &= -\log\left(\frac{\exp(x[class])}{\sum_j \exp(x[j])}\right) \\
&= -x[class] + \log\left(\sum_j \exp(x[j])\right)
\end{aligned}
\tag{1}
$$

where $x$ is a vector of dimension $J$ that the neural network inputs into the function with sample z, then outputs. The element value of this vector is between negative infinity and positive infinity, and the Softmax function normalizes this vector to between 0 and 1. *class* is the number of true classes of samples.

The cross-entropy loss function uses an inter-class competition mechanism, which is better at learning information between classes, but only cares about the accuracy of the prediction probability for the correct labels while ignoring the differences of other non-correct labels. It leads to a scattering of learned features [21]. Therefore, the optimization scheme demands considering the reasons for learning label prediction errors. The multi-category cross-entropy loss function optimization GCELoss adopted in this paper calculates as follows.

$$Loss = -x[class] +$$
$$\log\left(\exp(-x[class]) + \sum_j \exp(x[j])\right) \tag{2}$$

The optimization formula GCELoss adds an $\exp(-x[class])$ factor to the logarithmic part of the original multi-classification cross-entropy loss function formula. It makes the gradient of the new loss function higher than the gradient of the initial loss function, which facilitates network convergence [24].

### 2.2.3. Learning Rate Fixed Step Decay Strategy

Learning rate decay is a strategy to prevent the learning rate from being too large and swinging back and forth when converging to the global optimum. Therefore, we need to use a learning rate decay strategy that allows the learning rate to keep decreasing exponentially with the number of training rounds. The optimizer used in deep learning optimizes the learning rate, mainly during the backward operation process. The learning rate decay is a method of decaying per iteration, mainly during the forward operation process [11,25].

The idea of learning rate setting is to use a higher learning rate in the early stage to ensure faster convergence and a lower learning rate in the later stage to ensure stability. There are four main learning rate decay strategies: fixed-step decay, exponential decay, multi-step decay, and custom function decay [12]. Figure 3 shows the curves of the four strategies that dynamically adjust the learning rate as the iterations proceed.

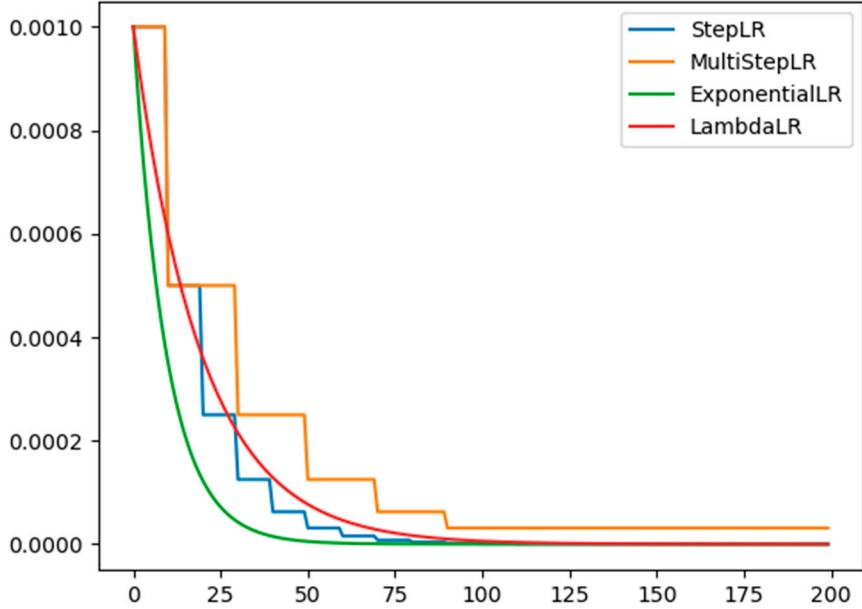

**Figure 3.** Four strategies for learning rate decay.

The learning rate is essential in neural networks, and the way to find the optimal learning rate is the key to how accurate the training. If the learning rate is too low, it will cause the loss rate of the network to decrease very slowly. If the learning rate is too high, the magnitude of the parameter updates is higher, which causes the network to converge to a local optimum point or causes the loss rate to increase directly [26].

In this paper, we use the learning rate fixed step decay strategy, which means that the learning rate decreases by a multiple after each fixed-step. Its calculation formula is as follows:

$$lr = lr_{initial} \times \tau^{(epoch/step)} \tag{3}$$

where $lr_{initial}$ is the initial learning rate, $\tau$ is the multiplier value, *epoch* is the number of training iterations, and *step* is the fixed step size.

The initial learning rate of the AlexNet network model is 0.001, and the learning rate decreases by 0.5 times every ten steps. While the initial learning rate of the improved AlexNet network model remains the same, and the learning rate chooses to decrease by 0.1 times every 50 steps. Because the decay of the learning rate of the original AlexNet will make the learning rate drop very slowly, making the training curve prone to loss value explosion and oscillation. After adjusting the parameters, the final learning rate will be in a suitable range, thus speeding up the convergence of the training curve. The calculation formulas are as follows:

$$lr1 = 0.001 \times 0.5^{(epoch/10)} \tag{4}$$

$$lr2 = 0.001 \times 0.1^{(epoch/50)} \tag{5}$$

Now, the two learning rate decay strategies are compared under the same initial learning rate with the Adam optimizer algorithm, as shown in Figure 4.

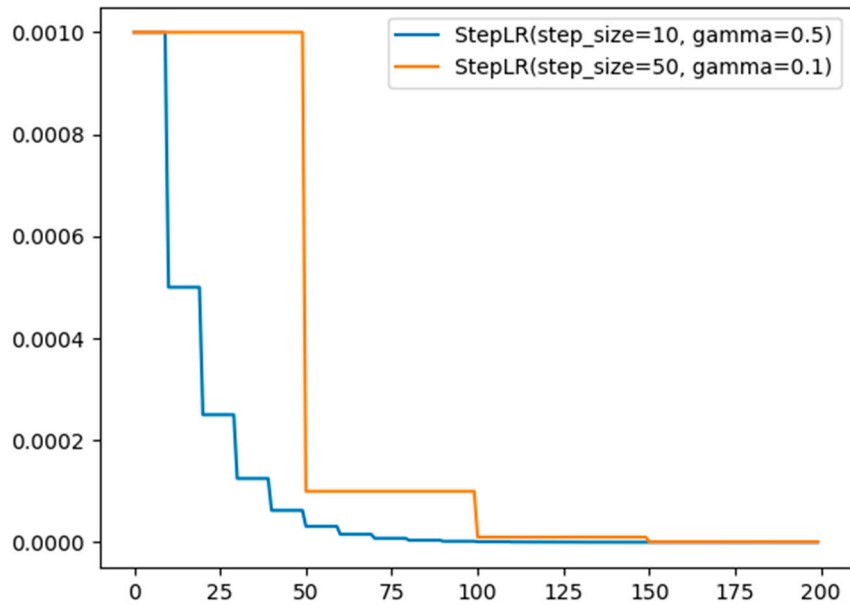

**Figure 4.** Comparison chart of two learning rate fixed step decay strategies.

We can see from Figure 4 that the fixed-step decay strategy of the learning rate after re-adjusting the parameters converges faster and facilitates the training of the network model.

### 2.3. Datasets and Experiment Scheme

To test the performance of the proposed improved AlexNet network model, we acquire a large number of Microcolumn Gel Card blood group images through the operating system of a blood group analysis instrument. Each Microcolumn gel card blood group image has

six microcolumn tubules. We split the tubules into six different Microcolumn Tubule blood group images for use in the experiment.

### 2.3.1. Dataset Description

The blood group images of the microcolumn gel cards are acquired through the operating system of the blood group analysis instrument, as shown in Figure 5. The six microcolumn tubules per microcolumn gel card, each a blood group image, are used as the data set for the training part of the blood group interpretation system.

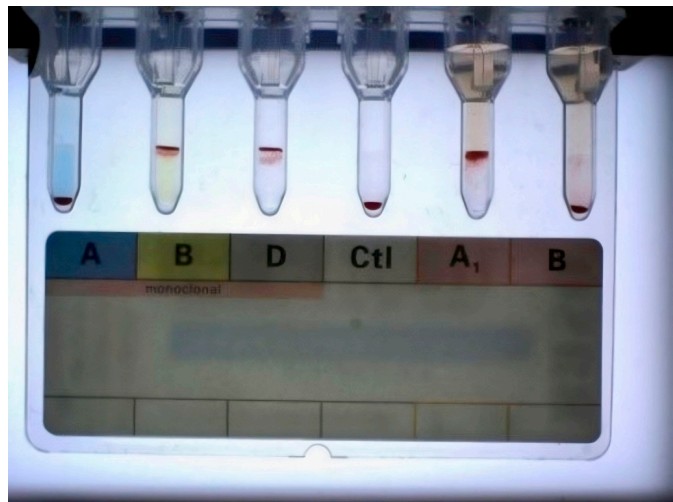

**Figure 5.** The microcolumn gel card blood group image.

Observe the characteristics of the erythrocyte agglutination reaction phenomenon in microcolumn tubules. According to the interpretation criteria of erythrocyte agglutination intensity based on the microcolumn gel detection technique [4,14], we can divide the microcolumn tubule blood group images into nine categories. They are negative (−), slightly positive (±), weakly positive (1+), moderately positive (2+), positive (3+), strongly positive (4+), the double cluster of cells (DP), hemolysis (H), and empty column (empty) [19]. The categories of the blood group images are shown in Figure 6.

The dataset images are cropped into images with $224 \times 224$ pixels to facilitate feature learning of the network model. The total number of blood group image datasets used in the blood group reading system is 837. They are divided into the training set, the validation set, and the test set with a ratio of 7:3:2, and the number of categories of the blood group image dataset is nine. The dataset is shown in Table 2.

**Table 2.** Dataset Division of Blood Group Reading System.

| Training Set | Validation Set | Test Set | Total Dataset |
|:---:|:---:|:---:|:---:|
| 479 | 197 | 161 | 837 |

Both the training and validation sets are in the model training phase [13]. The data in the training set is the object of neural network learning, and the validation set is used to show the approximate accuracy when the model trains to a certain step. When the model trains, the accuracy will be high and low. The last point of the training accuracy is not necessarily the highest value. The existence of the validation set can pick out the model parameters with the best accuracy. The test set tests the goodness of the trained model, whose accuracy is calculated from the confusion matrix generated by the model.

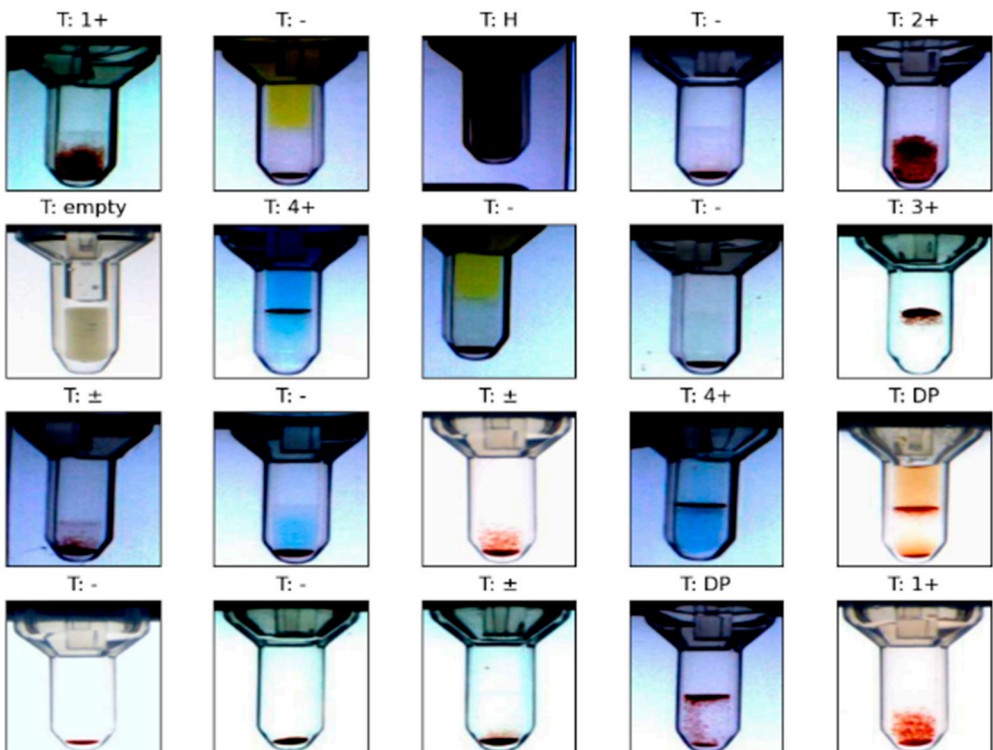

**Figure 6.** Nine categories of microcolumn tubule blood group image datasets.

### 2.3.2. Experiment Scheme

To demonstrate the advantages of the proposed pre-trained improved AlexNet network model step by step and explicitly, it is compared with the original AlexNet network structure [16], pre-trained AlexNet-Attention network structure [21], AlexNet + StepLR (50, 0.1), and AlexNet + GCELoss. Some frequent classification network models are also compared, such as VGG16 [27], ResNet5 [3], and DenseNet201. After reviewing a large amount of literature, they are compared with AlexNet + SVM [19], AlexNet + ELM [15], and other improved schemes.

They will experiment in some of the same training environments. The AlexNet network model trains with a batch size of 128, an optimizer of Adam, a weight decay value of 0.0005, a momentum of 0.9, a loss function with a multi-classification cross-entropy loss function, and an initial learning rate of 0.001. The learning rate decay strategy uses a fixed step decay function (StepLR), and the number of training iterations is 440.

The confusion matrix is a metric for judging the results of a model and is part of the model evaluation and is often used to evaluate the merits of a classification model. It summarizes the records in the data set in the form of a matrix according to two criteria. The two criteria are the real category and the category predicted by the classification model. The rows of the matrix symbolize the authentic values, and the columns of the matrix describe the predicted values [23,28–31]. The confusion matrix analysis is shown in Table 3.

**Table 3.** The Confusion Matrix Analysis Table.

| Predicted \ Actual | Positive | Negative |
|---|---|---|
| Positive | TP | FP |
| Negative | FN | TN |

Positive is a positive sample, and Negative is a negative sample. Each row of the confusion matrix corresponds to all samples predicted to belong to that class, and the diagonal line of the confusion matrix indicates the number of samples predicted correctly.



We hope that the prediction categories will be on the diagonal of the matrix when the network makes its prediction. The more densely the predictions are distributed on the diagonal, the better the model performance. The confusion matrix also makes it easy to see which categories the model are prone to classify incorrectly.

The accuracy, precision, recall, and specificity can be calculated by using the confusion matrix. Notice that the accuracy is for all samples, and the precision, recall, and specificity are for each category [12,19]. The formulas for their calculation are as follows:

$$Precision = TP/(TP + FP) \tag{6}$$

$$Recall = TP/(TP + FN) \tag{7}$$

$$Specificity = TN/(FP + TN) \tag{8}$$

$$Accuracy = (TP + TN)/(TP + FP + FN + TN) \tag{9}$$

*Precision* indicates the proportion of correct predictions among all Positive types predicted by the model. *Recall* indicates the proportion of correct model predictions among all true Positive types. *Specificity* indicates the proportion of all true Negative types for which the model predicts the correct Negative types. *Accuracy* indicates the proportion of the number of samples correctly classified by the model to the total number of samples (all categories). In general, the higher the precision, recall, and accuracy, the better the model [3].

## 3. Results

To evaluate the performance of the proposed improved AlexNet network model and compare it with the performance of traditional classification methods on the blood group dataset based on the microcolumn gel detection method. The results of the blood group classification accuracy are shown in Table 4.

**Table 4.** Accuracy Results of Network Models for Blood Group Classification Table.

| Blood Classification Network Model | Classification Accuracy (%) |
|:---:|:---:|
| AlexNet | 93.8 |
| AlexNet-Attention | 95.3 |
| VGG16 | 91.2 |
| ResNet50 | 93.4 |
| DenseNet201 | 88.1 |
| AlexNet + SVM | 94.3 |
| AlexNet + ELM | 92.9 |
| AlexNet + StepLR (50, 0.1) | 96.3 |
| AlexNet + GCELoss | 94.5 |
| SEBlock + AlexNet + StepLR (50, 0.1) + GCELoss | 96.9 |

To better demonstrate the performance of the proposed improved AlexNet network model on the blood group dataset, the training effect plots, confusion matrices, and data analysis tables for each category are compared between the original AlexNet network and the improved AlexNet network now.

### 3.1. Training Effect

The training effect plots of the original AlexNet network and the improved AlexNet network are shown in Figure 7a,b.

From Figure 7, we can see that after the training of the original AlexNet network in (a), the accuracy of the training set is 99.3% on average. The accuracy of the validation set is 91.7% on average, and the training time is 729.207 s. In comparison, after training the improved AlexNet network in (b), the accuracy of the training set is 99.5% on average and

is more stable. The accuracy of the validation set is 93.2% on average, and the training time is 756.978 s, which is within the affordable range. It demonstrates that, by comparing with the training effect of the original AlexNet network model, the loss rate of the validation set of the improved AlexNet network reduces. The up-and-down fluctuations of the loss rate are lower, the stability is relatively better, and the accuracy rate is effectively improved.

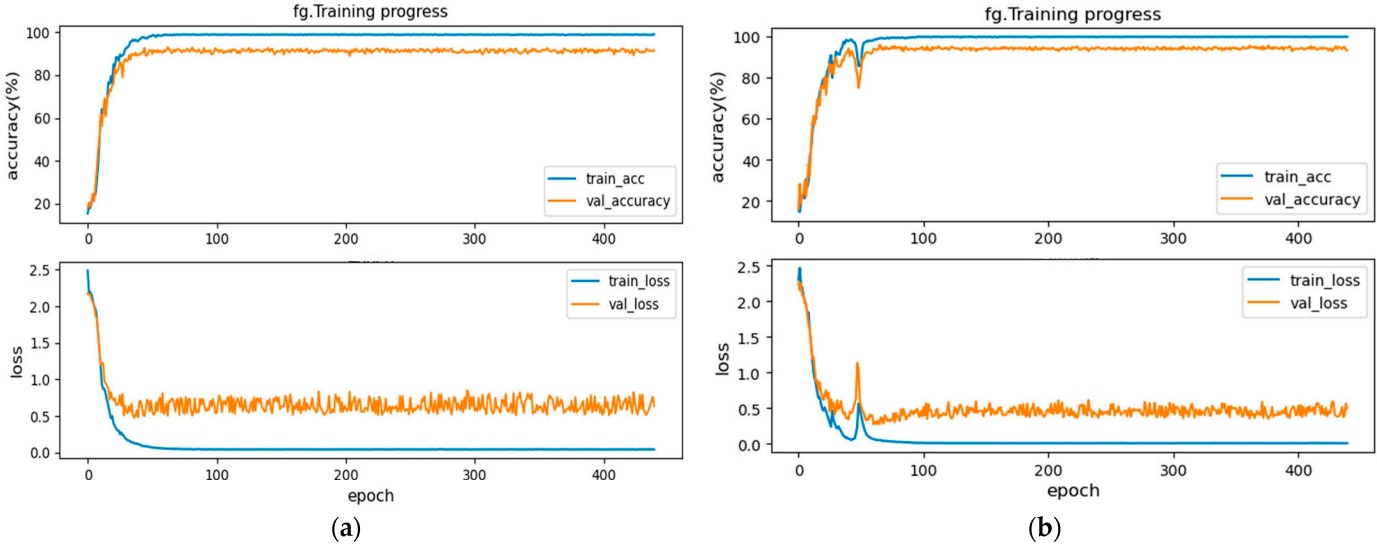

**Figure 7.** Comparison of the training graphs of the two network models. (**a**) Training curve of AlexNet network. (**b**) Training curve of improved AlexNet network.

### 3.2. Confusion Matrix Results

After loading the parameters of the trained network model, the confusion matrix of the original AlexNet network for the test set is shown in Figure 8a. The precision, recall, specificity, and accuracy are shown in Figure 8b.

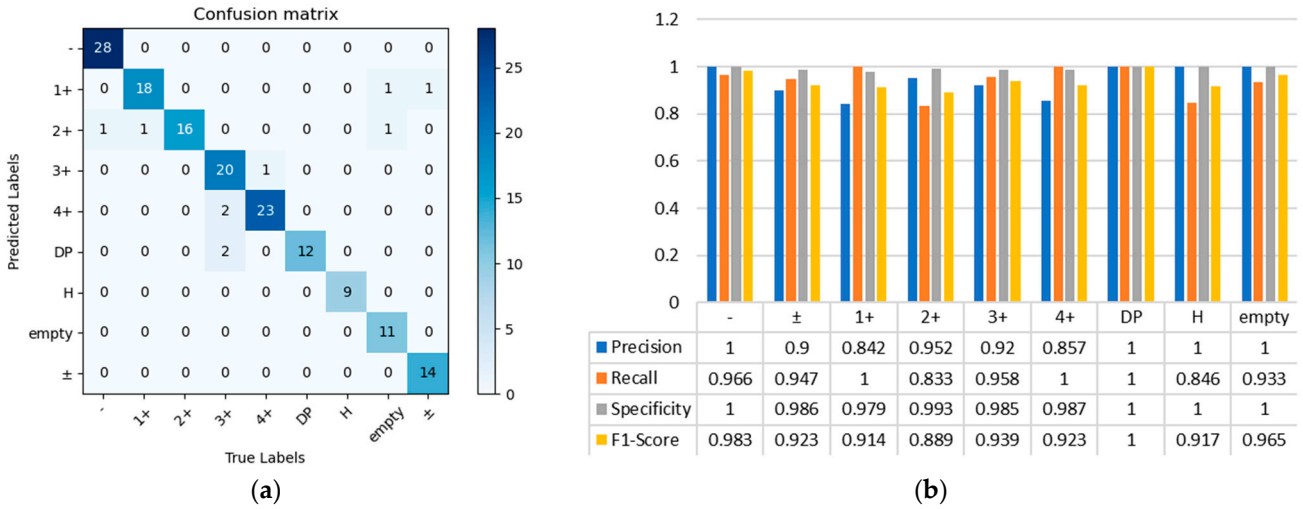

**Figure 8.** The original AlexNet network. (**a**) The confusion matrix. (**b**) Histogram of data analysis for each category.

The confusion matrix and the histogram of category data analysis show that the original AlexNet network training model loaded, excluded the blood type category with blurred boundaries, and showed multiple misclassifications, such as 3+ and DP, and empty and 1+. These categories are precisely the categories with low accuracy or recall. The

results are predicted in batches for the blood group images and visualize results. Through calculation, the accuracy of the model can reach 93.8%.

For comparison with the original network model, the parameters of the improved AlexNet network model load into the prediction program, and the test set confusion matrix is shown in Figure 9a. The precision, recall, specificity, and accuracy are shown in Figure 9b.

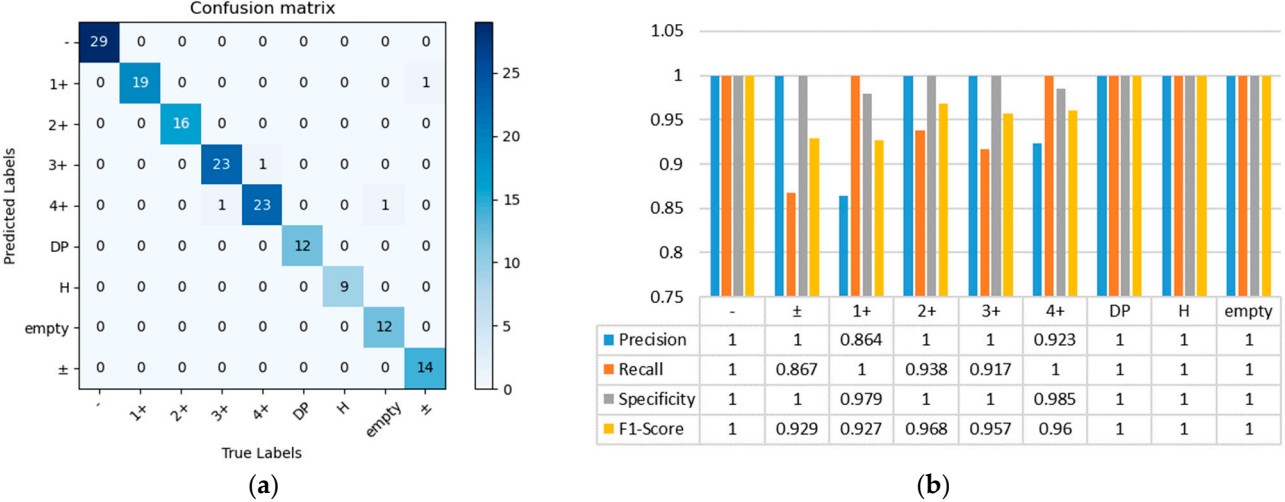

(**a**) (**b**)

**Figure 9.** The improved AlexNet network. (**a**) The confusion matrix. (**b**) Histogram of data analysis for each category.

We can see from the confusion matrix and the histogram of category data analysis that the number of errors in the test set will highly reduce after the improved AlexNet network training model is loaded. The errors mainly appear in blood group categories with blurred boundaries, such as between 4+ and 3+, ± and 1+. Their overall category accuracy and recall rates were generally higher, resulting in a substantial increase in the accuracy of the test set. Through calculation, the accuracy of the model can reach 96.9%. The results are predicted in batches for the blood group images and visualize results. The prediction results graph is shown in Figure 10.

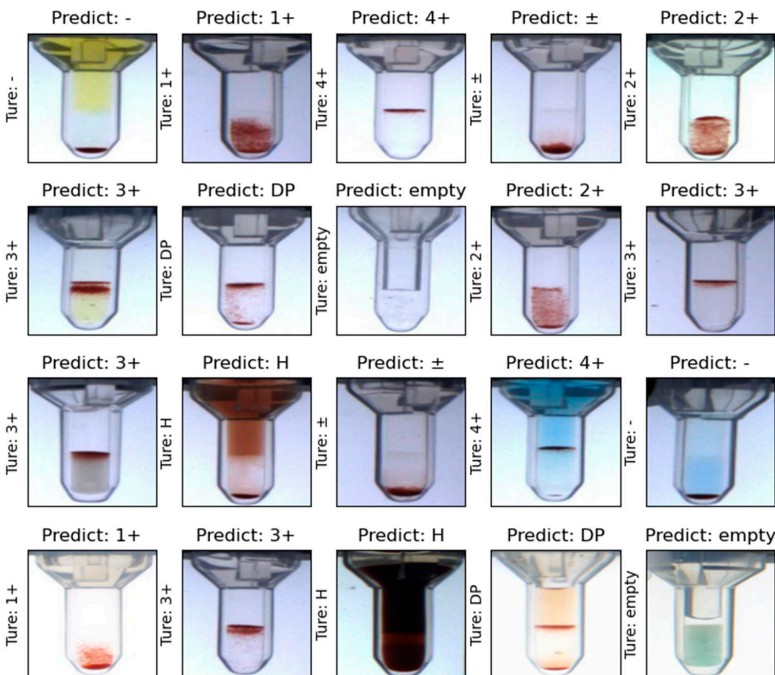

**Figure 10.** Prediction results of blood group images under the improved AlexNet model.

## 4. Discussion

From Table 3 and Figures 6–10, we can see that among the traditional classification network models, the DenseNet201 model has the worst accuracy because the features of microcolumn gel card image are complex and the calculation cost is high, while DenseNet201 model is suitable for the situation with few parameters and calculation cost. However, VGG16 and ResNet50 models are unstable due to the severe vibration of loss value in the training process. In contrast, the AlexNet model is more dominant in the field of blood type interpretation of microcolumn gel cards. Moreover, no matter the addition of attention mechanism, learning rate attenuation function tuning or loss function optimization algorithm can improve the accuracy of blood group classification. The improved AlexNet model obtained the best accuracy of blood group classification. Compared with other improved schemes in the AlexNet model, the addition of SVM and ELM limited the improvement of interpretation accuracy due to their slow learning rate and low generalization performance. However, the improved scheme of attention mechanism combined with loss function optimization and learning rate attenuation function tuning could better improve the classification performance of the AlexNet model in the field of blood group interpretation in microcolumn gel cards.

In summary, the comparison between the original and the improved network models demonstrates the feasibility of the improvement scheme in the field of blood group interpretation based on the microcolumn gel card, with the accuracy greatly improved.

## 5. Conclusions

In this paper, the improved AlexNet network structure, namely SeBlock + AlexNet + StepLR (50, 0.1) + GCELoss model, is proposed for use in a blood group interpretation system based on the microcolumn gel detection technique. By making the best of the attention mechanism, loss function, and learning rate decay function, we can further improve the performance of the AlexNet network. The improved AlexNet network model exhibits robust feature description capability for MGDT blood group images. The combination of the channel attention mechanism SEBlock takes into account the weight of each channel of the images, which helps feature distinct regions become more visible. The combination of loss function optimization GCELoss considers the difference of non-correct labels outside of inter-class information, which helps to focus the features more and reduce the error rate. The tuning of the learning rate decay function considers that the learning rate will make the training curve prone to loss-value explosion and oscillation when the decay is slow, which helps to train the model.

Through experiments, the proposed improved AlexNet network model outperforms the current AlexNet network model and the classification interpretation on the dataset based on blood group images of microcolumn gel cards. In our future work, we will explore the optimization of the loss function and the learning rate decay function in more CNN models. It enables the fully automated blood group analysis instrument to interpret the test results of microcolumn gel cards faster and more accurately.

**Author Contributions:** Conceptualization, R.S.; methodology, R.S.; software, R.S.; validation, R.S. and J.W.; formal analysis, R.S.; investigation, R.S.; resources, P.Z.; data curation, J.W.; writing—original draft preparation, R.S.; writing—review and editing, R.S.; visualization, R.S.; supervision, P.Z.; project administration, P.Z.; funding acquisition, P.Z. All authors have read and agreed to the published version of the manuscript.

**Funding:** This research was funded by the National Natural Science Foundation of China (61903050) and the Universities Natural Science Foundation of Jiangsu Province (20KJB120008).

**Data Availability Statement:** The data presented in this study are openly available in this article.

**Acknowledgments:** The authors gratefully acknowledge the financial support from the National Natural Science Foundation of China (61903050) and the Universities Natural Science Foundation of Jiangsu Province (20KJB120008).

**Conflicts of Interest:** The authors declare no conflict of interest.

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
