# Peer review of "Blood Group Interpretation Algorithm Based on Improved AlexNet"

_electronics, doi:10.3390/electronics12122608_

Round 1
Reviewer 1 Report
Title: Blood Group Interpretation Algorithm Based on Improved AlexNet
The topic is of interest and I enjoyed reading this article, however I have few major concerns and suggestions
Abstract:
1. The authors have mentioned about the background but however I feel they have some conclusive statements in the background
2. The abstract is written in future tense
3. The authors should mention at least 6-8 key words which are relevant to the topic
Introduction:
1. Introduction needs revisions, English editing must be considered
2. The authors have not followed any particular pattern of references, it is very inappropriate way. (reference starts with 12-14) Not according to MDPI guidelines
3. The authors should mention the values for high precision, high sensitivity in line 45
4. Introduction needs to be rewritten
Methodology:
1. Authors have written this section quite well, however why did they not consider comparing the models performance with the human experts/ specialists’?
Results: Results write up needs to elaborated
Discussion:
1. Discussion needs to elaborated, compared with previously reported models
2. How is this model superior to the previously reported models
3. Add note on the feasibility
4. Should report on the limitations of this study
5. Should add a note on policy making on marketing and applicability in hospital settings
References:
1. Is not a correct way of presentation
2. Not according to MDPI guidelines

Author Response
I have Modified as required

Reviewer 2 Report
Blood Group Interpretation Algorithm Based on Improved
AlexNet
==================
The paper proposed to use Alexnet model with attention information for the blood group interpretation.
The paper is interesting and has significant application.
However, the paper requires following issues to be fixed before acceptance.
1. The paper lacks open questions. Eg what are the limitations of the existing methods?
2. What is the significance of the proposed work?
3. What is original network in the abstract? It is confusing.
4. The abstract is jumbled with less coherent sentences. Please rewrite it and maintain the coherency.
5. Highlight the contributions in the introduction.
6. Discuss the recent literature works as shown below related to the digital health and deep learning.
https://www.nature.com/articles/s41598-021-03287-8
7. CBAM attention is also popular in digital health. Please explain the following paper that uses CBAM attention.
https://link.springer.com/article/10.1007/s10489-020-02055-x
8. Please remove step lr and loss from the contribution as they are normal and fundamental thing in the deep learning.
9. Please compare the proposed method with the state of the art methods. Currently, the comparison is just with few pre-trained DL models.
Author Response
I have modified as required.

Reviewer 3 Report
The manuscript received is of considerable quality, presenting promising findings from the authors. However, there is a need for heightened ethical consideration, particularly regarding trust in results produced by artificial intelligence in the context of medicine. It is vital that the authors specify how they intend to interpret these results from the perspective of the end user.
The authors are advised to take into account the following comments:
1. The expert suggests that the content of section "2.1. The AlexNet Architecture" be relocated to section "1. Introduction" or a newly established subsection, namely "Related works."
2. The majority of the information from "2.2.1. The Channel Attention Mechanism SEBlock" should likewise be shifted to "1. Introduction" or a separate section titled "Related works."
3. The bulk of the content from "2.2.2. The Multi-categorical Cross-entropy Loss Function Algorithm Optimization" is also recommended to be moved to "1. Introduction" or a separate "Related works" section.
4. Some of the information from "2.2.3. Learning Rate Fixed Step Decay Strategy" should also be transferred to "1. Introduction" or a separate "Related works" section.
The underlying message in comments 1-4 is to separate established methodologies from the "2. Materials and Methods" section. The authors should only keep those in Section 2 that offer enhancements to existing solutions.
5. The "4. Discussion" section should enumerate the limitations of the suggested method. The authors should also analyze and discuss the enhancements that had a substantial impact on the final outcome.
6. Given the sensitive nature of the subject field (medicine), it is crucial to highlight the clarity and interpretability of the achieved results. Simply achieving impressive classification results is insufficient- mechanisms must be in place to instill trust in the decision-making process.
7. The conclusion section should provide more detail, particularly about i) the quantitative results achieved in the study and ii) the limitations of the proposed methodology.
On the whole, this manuscript could be accepted pending some minor amendments.
Author Response
I have modified as required.

Round 2
Reviewer 1 Report
Kindly write the abstract in past tense, line 11 says "role of deep learning in blood group interpretation system will be studied"
Over all the corrections are acceptable
Kindly write the abstract in past tense, line 11 says "role of deep learning in blood group interpretation system will be studied"
Over all the corrections are acceptable
Author Response
The abstract has been changed to the past tense.

Reviewer 2 Report
Thanks for the revision; however, comments #6, 7, and 9 have not been addressed seriously as suggested.
Author Response
I have discussed and explained the first round of comments #6, 7 and 9.
